# Structure of a mitochondrial ribosome with fragmented rRNA in complex with membrane-targeting elements

Victor Tobiasson [1], Ieva Berzina [2] & Alexey Amunts [1] ✉

Mitoribosomes of green algae display a great structural divergence from their tracheophyte relatives, with fragmentation of both rRNA and proteins as a defining feature. Here, we report a 2.9 Å resolution structure of the mitoribosome from the alga *Polytomella magna* harbouring a reduced rRNA split into 13 fragments. We found that the rRNA contains a non-canonical reduced form of the 5S, as well as a permutation of the LSU domain I. The mt-5S rRNA is stabilised by mL40 that is also found in mitoribosomes lacking the 5S, which suggests an evolutionary pathway. Through comparison to other ribosomes with fragmented rRNAs, we observe that the pattern is shared across large evolutionary distances, and between cellular compartments, indicating an evolutionary convergence and supporting the concept of a primordial fragmented ribosome. On the protein level, eleven peripherally associated HEAT-repeat proteins are involved in the binding of 3′ rRNA termini, and the structure features a prominent pseudo-trimer of one of them (mL116). Finally, in the exit tunnel, mL128 constricts the tunnel width of the vestibular area, and mL105, a homolog of a membrane targeting component mediates contacts with an inner membrane bound insertase. Together, the structural analysis provides insight into the evolution of the ribosomal machinery in mitochondria.

Ribosomes consist of the small subunit (SSU) that selects aminoacyl-tRNAs cognate to mRNA codons, and the large subunit (LSU) that contains the peptidyl transferase centre and polypeptide exit tunnel. Both subunits are composed of a catalytic rRNA core and scaffolding proteins. In mitochondria, the rRNA and the associated protein architecture has diverged considerably relative to their bacterial ancestors, which allowed an acquisition of accessory functions[1–6]. This diversification is partially driven by the continuous reduction of the mitochondrial genome, resulting in the generally reduced size of mitoribosomal RNAs[7,8]. Previously published mitoribosomal structures mostly contained subunits consisting of a single or two rRNA fragments[1–4,8–12]. The discovery of the mitochondrial DNA from the chlorophycean green alga *Chlamydamonas reinhardtii* that encodes discontinuous rRNA genes split into 12 modules, led to a suggestion that the primordial ribosome might have consisted of a complex of individual small RNAs contributing specific functional elements[13]. Indeed, a protoribosome consisting of two universal internal RNA fragments of only ~70 nucleotides each is capable of mediating peptide-bond formation and displays a limited catalytic ability[14], thus supporting the hypothesis that a modern ribosome might have evolved from smaller ancestral fragments. Bioinformatic studies have also reported a more progressive case of mitochondrial genome fragmentation in malaria-causing parasites from the phylum Apicomplexa, where mitoribosomal RNA is heavily reduced[15]. However, the lack of structural data leaves open the questions of the evolutionary construction of rRNA. Moreover, it remains unclear if any factors

[1]Science for Life Laboratory, Department of Biochemistry and Biophysics, Stockholm University, 17165 Solna, Sweden. [2]Department of Medical Biochemistry and Biophysics, Karolinska Institute, 17177 Stockholm, Sweden. ✉e-mail: amunts@scilifelab.se

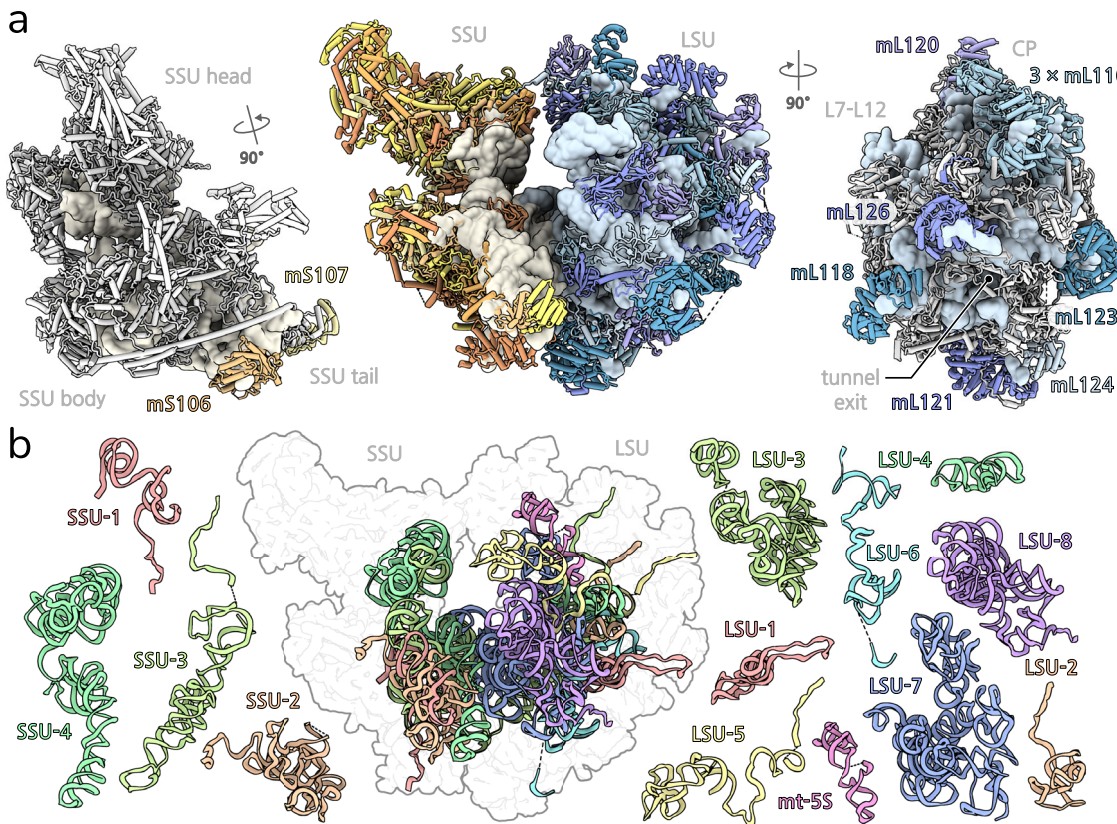

**Fig. 1 | Overall structure of the *P. magna* mitoribosome with fragmented rRNA.** **a** View of the monosome with proteins coloured by subunit, and individual SSU (left) and LSU (right) with their corresponding HEAT proteins highlighted. **b** Arrangement of assembled rRNA fragments together with their individual structures.

facilitate the stabilization of individual rRNA fragments into a single functional unit and how the 5S rRNA could be removed from most of the known mitoribosomal structures.

To better understand the evolutionary development of the ribosome, a comparative analysis of mitoribosomes with fragmented rRNA coupled with phylogenetic inferences is instrumental. Indeed, a recent report of *C. reinhardtii* mitoribosomal subunits revealed divergence in some of the key building blocks[16], however the detailed architecture remained elusive. Here, we characterised intact mitoribosomes from the chlorophycean, colourless green alga *Polytomella magna*[17], which is assumed to have distinct respiratory chain complexes similar to other studied *Polytomella* species[18,19]. The *P. magna* mitochondrial genome is linear and made up entirely of palindrome coding regions, with every gene followed by an inverted tandem duplication[20]. It has 12 different rRNA transcripts that have been modelled into potential SSU and LSU rRNA secondary structures[21].

## Results
### Cryo-EM structure of the *P. magna* mitoribosome
We purified mitoribosomes from cultured *P. magna* cells and resolved the cryo-EM structure of the monosome to an overall resolution of 2.9 Å (Supplementary Table 1). Further data processing and classification revealed distinct classical and hybrid tRNA binding states, supporting the engagement in active translation (Supplementary Figs. 1, 2). Protein sequence identification was carried out using a combination of cryo-EM density interpretation and transcriptome data, which we assembled de novo using reads retrieved from the Sequence Read Archive under the accession code SRX710730.

Our model of the mitoribosome contains 13 rRNA fragments (LSU1-8, SSU1-4, mt-5S), which lengths ranging from 73 to 576 nucleotides, as well as 94 proteins, including 26 newly identified (Fig. 1,

Supplementary Fig. 3, Supplementary Table 2). All bacterial protein homologues are conserved with the exception of the universally lost bS20, and the functional sites for mRNA and tRNA binding remain largely unaltered (Supplementary Fig. 3). In addition to fragmentation of the rRNA, we also noted three occurrences of protein fragmentation in the conserved bacterial homologues uS3m, uS4m and uS7m, each one of which is split into two nuclear-encoded fragments (Fig. 2b). Most of the remaining mitoribosomal proteins are extended compared to the bacterial homologues. Together, those extensions sum to ~1500 amino acids. The extensions of uS10m, mS31 as well as uS4m and mS45 provide the core of two protuberances present on the back of the SSU head and body (Fig. 2a, Supplementary Figs. 3, 4a). The accumulation of protein mass on the surface can be explained by rapid mutation rates observed for genes coding for mitochondrial proteins in Chlorophycea[22], in conjunction with a hydrophobic ratcheting mechanism[23].

### HEAT-repeat proteins bind 3′ rRNA termini
We modelled eleven HEAT proteins in the structure, seven of which bind specifically to the 3′ termini of rRNA fragments (Fig. 1b, Supplementary Figs. 3, 4 and 5). Although, in mammals, proteins containing other RNA-binding motifs, such as the helical pentatricopeptide repeat that forms antiparallel α-helices arranged in a superhelix, have been reported to regulate mitoribosomes in a modular fashion[24], we do not observe presence of other than HEAT repeats in the structure. At the central protuberance, three HEAT-repeat proteins, all copies of mL116 (mL116-1, mL116-2, mL116-3), come together to form a distinct trimer of ~120 kDa. Each copy of mL116 consists of 21 helices and anchored via the N-terminus to the central protuberance (Fig. 3a). The trimer only displays a pseudo-symmetry, and the protein mL116-1 is positioned in a way that forms a helical bundle with mL116-3 and weak interactions

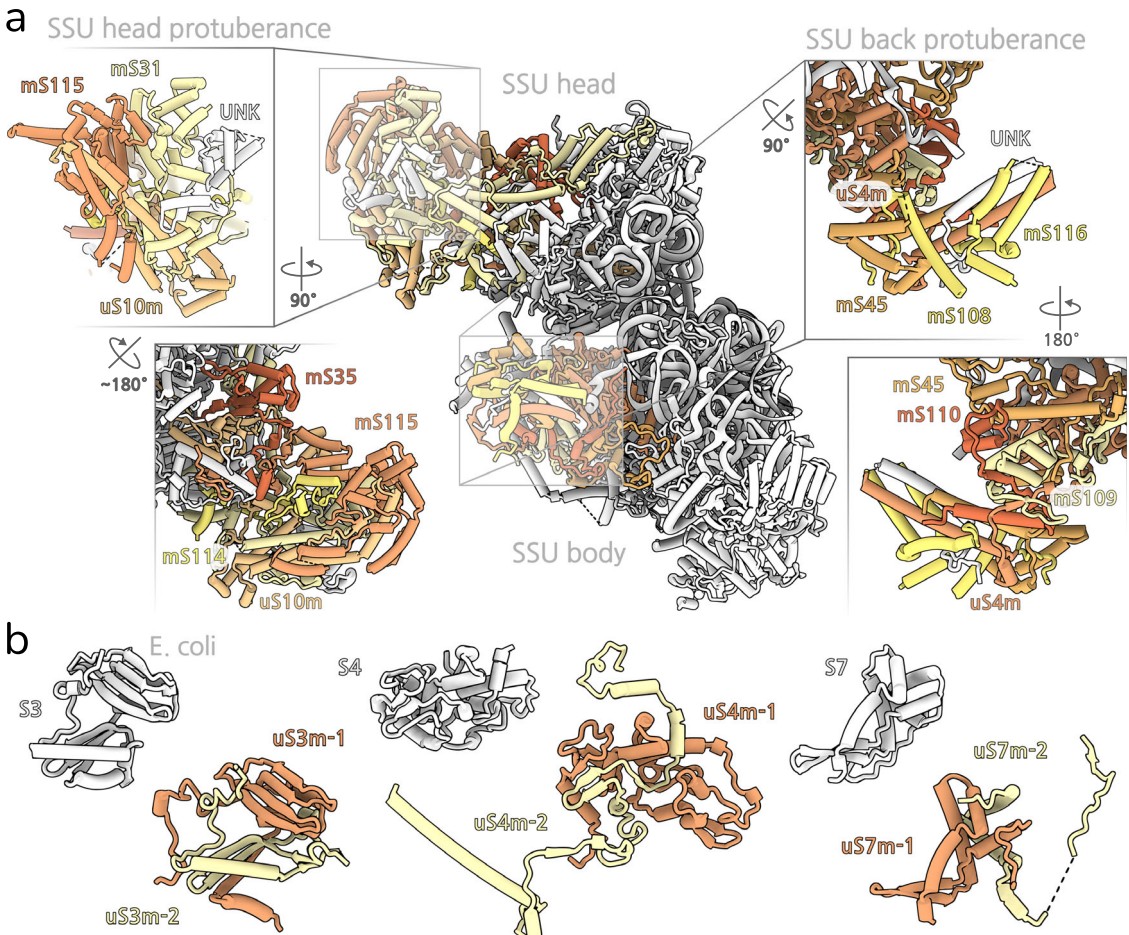

**Fig. 2 | SSU architecture and protein splitting. a** The head protuberance (left) is built upon an extension of conserved uS10m that serves as a platform for newly bound mS114 and mS115. The body protuberance (right) is built upon an extension of uS4m that serves as a platform for newly bound mS109 and mS110. **b** Each one of

the fragmented mitoribosomal proteins uS3m, uS4m, uS7m consists of two separate proteins (distinct colours for individual proteins) encoded in the nuclear genomes. Comparison with *E. coli* counterparts (white) illustrates the protein splitting phenomenon.

with mL116-2. There are no direct contacts between mL116-2 and mL116-3. With respect to rRNA, each protein copy binds a 3′ terminus of a different fragment: LSU-2, LSU-3, and LSU-5. The individual subunits are recruited to a common motif GXXAAA (Fig. 3b) present in each of the three rRNA termini. The shared motif together with the co-localised fragment termini thus rationalises the presence of the trimer.

### rRNA fragmentation patterns are conserved across large evolutionary distances

Based on our model, we constructed an rRNA secondary structure diagram and found that all the rRNA domains in the LSU and SSU are reduced compared to bacteria, resulting in a total rRNA length of ~1670 and ~1060 nucleotides, respectively (Supplementary Fig. 6). In addition, the fragmentation coincides with rRNA regions that are generally reduced in mitoribosomes of other species.

Comparing ours and the algal *C. reinhardtii*[16] structures with the bioinformatic prediction of the *P. falciparum* rRNA[15] reveals a similar fragmentation pattern, although the fragmentation appears to be less extreme in algae (Fig. 4, Supplementary Fig. 7). We observe colocalization of rRNA fragments across all domains of the LSU (with the exception of domain I which is predicted to be lost in *P. falciparum*) and across all domains of the SSU. We then compared the structural data with cytoplasmic ribosomes of *Euglena gracilis*[25] and kinetoplastids[26,27], where rRNA fragmentation is known to occur, and observed the same general pattern (Fig. 4).

One possible explanation of the convergence of rRNA fragmentation patterns relates to the set of structural constraints on folding and assembly that is intrinsic to the basic ribosomal architecture. In addition, the noncontiguous arrangement of rRNA fragments in the mitochondrial genome of *P. magna* might suggest a ribosome biogenesis pathway which is compatible with extensive fragmentation and does not require strict cotranscriptional folding[28]. Our data shows that most of the neighbouring fragments are directly associated with each other in the structure via base-pairing between their corresponding termini, and no additional protein linkers are found in the structure. Although, the fragmented rRNA of contemporary ribosomal subunits most likely evolved from ancestors with covalently continuous rRNA, the convergence of the rRNA fragmentation presents the idea that the pattern constitutes a record of ribosomal history from an early evolution of the translational system, as previously hypothesized[13].

### Evolutionary aspects of fragments LSU-1 and 5S rRNA

Although the overall rRNA core is similar to that of bacteria, we identified a distinct fragment LSU-1 that represents a truncated and permuted version of the bacterial rRNA H2-4 with the conserved branching in H23 (Fig. 5a). The circular permutation is achieved through fusing the 5′ and 3′ termini into a new internal loop upstream of H2, and new termini formed downstream of H4. LSU-1 is further bound by two HEAT proteins, mL126 and mL123 that scaffold the internal loop and the extended 3′ terminus, respectively. This

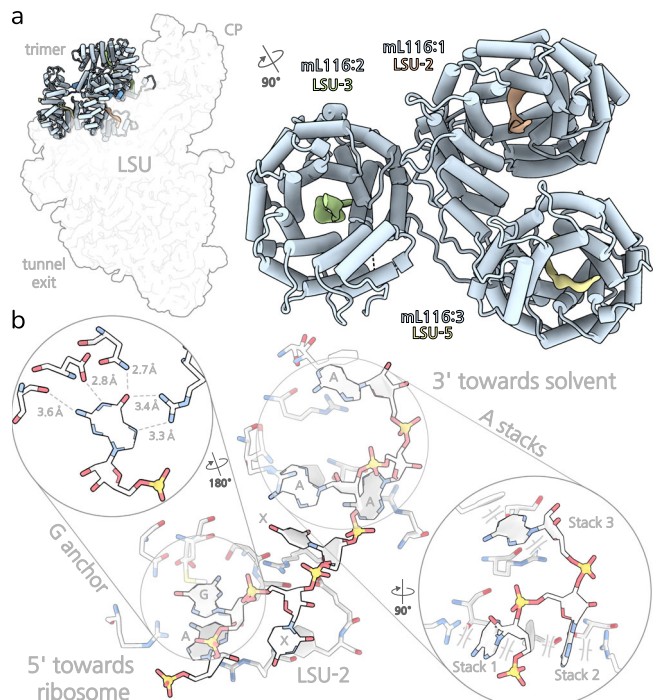

**Fig. 3 | mL116 binding to rRNA motif in fragments LSU-2, LSU-3, and LSU-5. a** mL116 trimer is found in the central protuberance (CP). Each copy binds a 3′ terminus of rRNA fragment (LSU-2, LSU-3, LSU-5) in its central groove. **b** Each one of the three mL116 monomers in the central protuberance of the LSU binds a 3′ terminus of rRNA via the motif GXXAAA. A specific guanine binding site followed by three adenine bases stacked between arginine and phenylalanine. Insets showing contact distances and specific guanine binding as well as alternative stacking view.

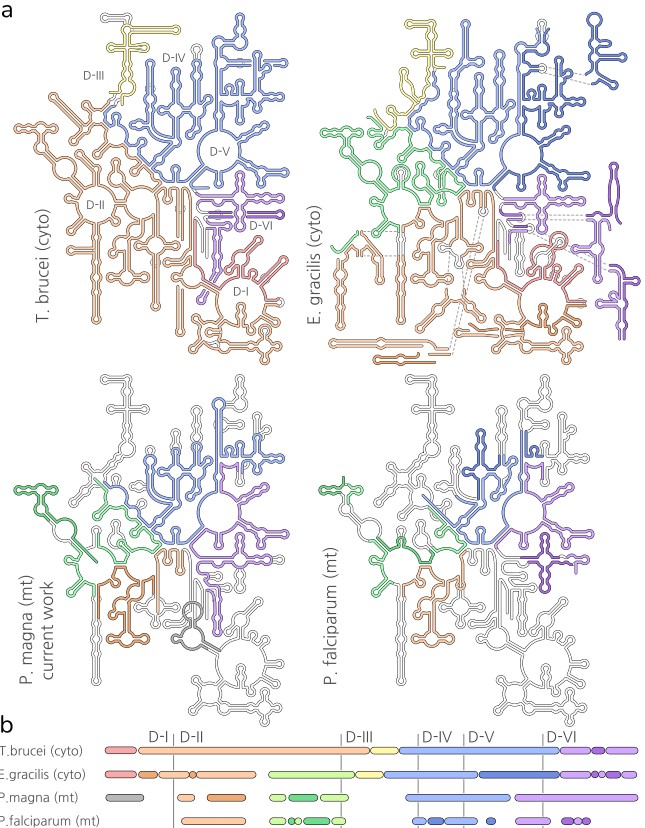

**Fig. 4 | LSU rRNA fragmentation patterns. a** Comparison of mitochondrial LSU rRNA from *P. magna* and *P. falciparum* and cytosolic LSU rRNA from *T. brucei* (PDBID: 4V8M) and *E. gracilis* (PDBID: 6ZJ3) highlights similar fragmentation patterns. Fragments spanning equivalent regions of the rRNA secondary structures coloured by hue, individual fragments highlighted by colour brightness. *E.coli* reference diagram as a white background. **b** Linear representation of the fragments' span.

arrangement is likely the result of a previous genomic tandem duplication of the LSU-1 rRNA gene, subsequent reduction of which would rearrange the helical ordering in the linear sequence and result in the formation of a circular permutation. This model of topological inversion has previously been suggested for proteins[29], and here we show that it can also be applied to rRNA, thus illustrating the plasticity of the ribosomal structure and the tight link between the evolution of the miotochondrial genome and the mitoribosome.

A fragment of a particular interest is the mt-5S rRNA. While 5S rRNAs are ubiquitous components of bacterial[30], archaeal[31], chloroplast[32,33], and eukaryotic cytosolic[34] ribosomes, they are predominantly absent from ribosomes of mitochondria, and have only been modelled in *C. reinhardtii*[16] and *Arabidopsis thaliana*[3]. Additional forms of truncated and altered 5S rRNA-like transcripts have previously been reported in the mitochondrial genomes across Eukarya[35]. In our structure, a mt-5S rRNA could be assigned based on the conserved domain γ. However, the identified rRNA is non-canonical and found in a shortened form, completely missing the domain α, and with a truncated domain β that lacks H2 and H3 (Fig. 5b). Although we couldn't identify a mitochondrial gene corresponding to the 5S rRNA in *P. magna*, it has been reported for *C. reinhardtii*[16]. This minimal mt-5S rRNA is positioned on the central protuberance by three proteins: uL18m, bl25m, and mL40 (Fig. 5c). While uL18m and bL25m bind domains α and γ, the mitochondria-specific mL40 caps the apex of the truncated domain β. With the deletion of domain α, uL18m has lost its binding interface to the rRNA but remains peripherally attached due to stabilising contacts with mL40. The newly formed 3′ and 5′ termini of the mt-5S rRNA are present at the three-helix junction which is further stabilized by the HEAT protein mL120.

The finding of the mitochondria-specific mL40 in our structure coupled to the truncated 5S rRNA provides an evolutionary insight. Particularly, mL40 is found in mitoribosomes of ciliates[4] and fungi[11] that lack the 5S rRNA, as well as of mammals where the rRNA has been replaced by a structural tRNA component[36,37]. This suggests that mL40 has been acquired by an ancestral mitoribosome before the loss of the mt-5S rRNA, and it has remained as a structurally important element. Consequently, a gradual reduction and removal of the 5S has been scaffolded by uL18m and mL40, which allowed other locally abundant components, such as the tRNA to bind to sites with pre-established affinity to RNA, thereby promoting the structural divergence of mitoribosomes[8].

### Evolutionary intermediates in the exit tunnel

During refinement of the cryo-EM map, we observed an additional continuous density extending ~60 Å outside the tunnel exit. The density correlates with the inner membrane plane reported by tomography of *C. reinhardtii* mitoribosomal complexes[16] and is associated with the mitoribosomal protein mL105 at the tunnel exit (Fig. 6a). Based on the size and positioning, it likely corresponds to the OXA1L insertase[38] (Supplementary Fig. 8). We previously found that in the ciliate *Tetrahymena thermophila*, mL105 is a homologue of the bacterial mobile signal recognition particle (SRP) binding protein Ffh[4]. In the current structure, although most of the rRNA that forms the binding site for Ffh in bacteria has been lost, the position of mL105 remains similar due to newly established interactions with protein extensions of bL24m and uL29m. This organisation stabilises the binding of the structurally conserved mL105, which suggests a functional importance. Comparison with the human mitoribosome-OXA1L complex[38] shows that mL105 would overlap with the mammalian

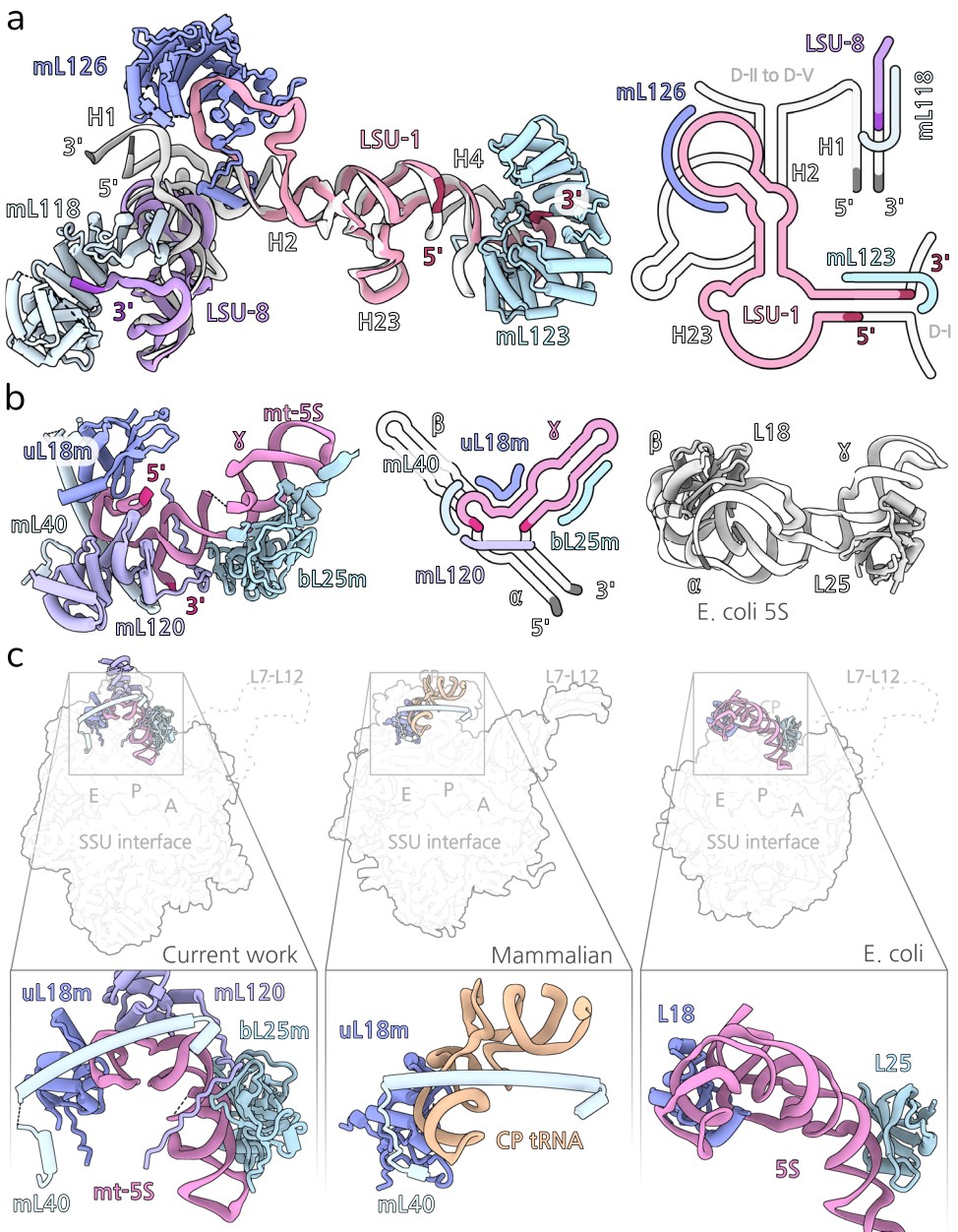

**Fig. 5 | Reduced LSU-1 and mt-5S rRNAs. a** LSU-1 fragment (pink) is bound by three HEAT protein superimposed with *E. coli* rRNA (white, PDBID: 7K00). A secondary structure diagram is shown on the right, and protein binding regions are indicated. **b** The mt-5S rRNA compared with *E. coli* 5S and their bound proteins. A secondary structure diagram is shown in the middle, and protein binding regions are indicated. **c** View from the interface of the mammalian mt-LSU of *P. magna*, *H. sapiens* (PDBID: 6ZM5) and *E. coli* shows the arrangement of the 5S rRNA in comparison to the structural CP tRNA. The mitochondria-specific protein mL40 is found regardless of the identity of rRNA in that region.

mL45, implying a similar membrane-mediating function. Thus, our structure suggests that the metazoa-specific mL45 has been recruited following the loss of mL105 and possibly compensates for missing membrane contacts following the removal of the ancestral SRP mediating protein.

Finally, in the vestibular area of the tunnel, we found a small 24-aa long helical protein mL128 that constricts the path and decreases the tunnel to a minimum width of ~11 Å (Fig. 6a). Its size, position and binding mode are similar to the human alpha-helix α0 of mL45 (Fig. 6b). mL128 is anchored from the opposite sides of the tunnel through uL22m and a reduced H50 rRNA. Molecular dynamics simulations in the human mitoribosome have shown that a mitoribosomal protein moiety in the vestibular area that induces a specific geometry would affect cotranslational protein folding by

restricting helix formation within the tunnel[38]. Therefore, mL128 might represent a solution to maintain this feature of the mitoribosomal exit tunnel.

## Discussion

In summary, our study illustrates that the structure of the algal mitoribosome can be used as a model to study rRNA fragmentation, a crucial yet a poorly understood aspect of ribosomal evolution. The rRNA being responsible for the structural core is a central theme of ribosomal evolution, and it has been continuously modulated[39], thus an open question in the field is how the catalytic rRNA has acquired its current size and complexity[40]. Our comparative analysis finds a conservation of the fragmentation pattern and supports the notion that it might represent an evolutionarily

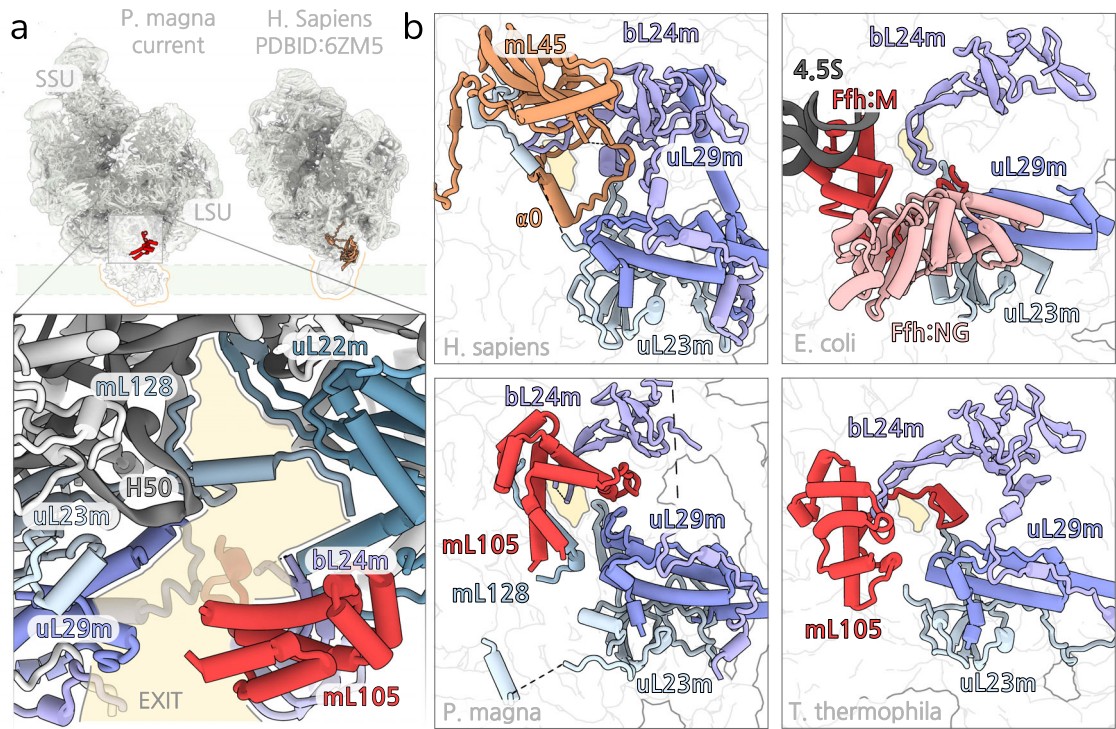

**Fig. 6 | Polypeptide exit tunnel and membrane contacts. a** Comparison of the density maps between *P. magna* and *H. Sapiens* mitoribosomes. The maps were filtered to 10 Å and aligned on the LSU. mL105 (*P. magna*) shown in red. Zoomed panels illustrate the position of mL105 at the tunnel exit. mL128 anchored constricts the path. **b** Comparison of the tunnel exit between *P. magna, H. Sapiens,*

*T. thermophila*, and *E.coli* shows that mL105 (*P. magna, T. thermophila*), mL45 (*H. Sapiens*), and Ffh M-domain (*E. coli*) occupy similar positions in different ribosomes. In addition, relative positions of mL128 (*P. magna*) and mL45-α0 (*H. sapiens*) are similar. The exit regions are indicated with a yellow background.

primitive feature of rRNA genes. Yet it remains to be seen whether a primordial ribosome had its rRNAs in pieces. Of the fragments observed in our structure, the permuted LSU-1 and the reduced 5S highlight different aspects of the ribosome variation and evolution. The possibility of rRNA domain permutations, as observed for LSU-1, couples the dynamics of genome evolution to that of the mitoribosome. The 5S rRNA that is incorporated into the central protuberance with mitochondria-specific protein mL40 represents an evolutionary intermediate between bacteria and mammalian mitoribosome, which adds a missing piece to the puzzle of our understanding of how the 5S rRNA can be removed and replaced without affecting the ribosomal structure. Finally, the observation of the membrane interfacing protein mL105 that is related to bacterial Ffh supports its role in mediating membrane contacts with a mitochondrial insertase and illustrates an intermediate of mitoribosomal evolution with respect to co-translational protein insertion.

## Methods
### Purification of mitoribosome
*P. magna* strain 63-9 was obtained from the SAG Culture Collection, University of Göttingen. Cells were grown at a room temperature (-22 C) without shaking in a medium consisting of 0.1% Tryptone, 0.2% Yeast Extract and 0.2% Sodium Acetate at pH 6.8. Cells were then propagated by repeated expansion at a late log phase ($8 \times 10^5$ cells/mL) until -27 L of culture were obtained. For the final cultures 2 L glass erlenmeyers with 700 mL media were used to allow aeration. The final culture was harvested at logarithmic growth ($4 \times 10^5$ cells/ml) by centrifugation at 1300 g, and the cell pellet was gently resuspended with homogenisation buffer (20 mM HEPES KOH pH 7.5, 300 mM mannitol, 5 mM EDTA) to wash out residual media. The harvested cells were resuspended 1:1 (w/v) in homogenisation buffer and lysed manually by

30 strokes of a 125 mL glass homogenizer fitted with a teflon plunger. The lysate was clarified by two rounds of centrifugation at 1300 g for 10 min. Unbroken cells pelted during clarification were lysed by another 30 strokes in the homogenizer. Mitochondria were isolated from the clarified lysates by centrifugation twice at 7000 g for 20 min. Mitochondria-rich pellet was resuspended in minimal volume of SEM buffer (20 mM HEPES KOH pH 7.5, 1 mM EDTA) and transferred onto two discontinuous sucrose gradients (15%, 23% 32% and 60% sucrose). The gradients were centrifuged at 72000 g for 60 min and mitochondria were extracted from the interface between the 32% and 60% layer using a pasteur pipette. Isolated mitochondria were flash frozen in liquid nitrogen.

All subsequent steps were carried out at 4 C. 20 mg of mitochondrial protein (as estimated by BCA assay of the isolated mitochondrial suspension) were lysed in 35 mL lysis buffer (25 mM HEPES KOH pH 7.5, 50 mM KCl, 15 mM MgOAc, 3.5% digitonin, 1 cOmplete[tm] EDTA-free Protease Inhibitor tablet) and stirred on ice. Insoluble mitochondrial material was pelleted twice at 30,000 kg and the clarified supernatant layered on 15 mL 34% sucrose cushion (25 mM HEPES KOH pH 7.5, 50 mM KCl, 15 mM MgOAc, 1 M sucrose, 1% digitonin). The sucrose cushions were centrifuged at 235,000 g for 3 h and the resulting proteinaceous pellets were resuspended in the same buffer without sucrose. The resuspended pellets were clarified twice by centrifugation at 12,000 g, and transferred onto a 15–30% linear sucrose gradient supplemented with 0.05% digitonin and centrifuged at 92,000 g for 90 min in rotor TLS-55. The gradient was fractionated by manually puncturing a hole at the bottom of the polycarbonate tube using a needle. The presence of mitoribosomes was gauged by the absorbance at 260 and 280 nm. Fractions with enriched 260 nm absorbance were pooled and buffer exchanged in centrifugal concentrators. The final sample was diluted to $OD_{260} = 1.8$.

### Cryo-EM data collection and processing

3 uL of purified mitoribosomes were applied to copper Quantifoil 400 R 2/2 grids precoated in house with ~3 nm continuous carbon. The sample was vitrified using a FEI Vitrobot mKIV with 3 s blotting at 4 C with the chamber at 100% humidity. Cryo-EM data was collected at SciLifeLab on a Titan Krios fitted with a Gatan K3 camera. The total fluence was 30 e⁻/Å² fractionated across 20 frames taken over a period of 2 s at pixel size of 0.864 Å/px on the detector level with a 70 μm aperture inserted. The images were collected with EPU 1.9 up to a nominal defocus of −2 μm. Beam induced motion was corrected with Motioncor2[41] and the parameters of the CTF initially estimated using GCTF[42] in RELION 3.1[43]. 45850 micrographs were selected for further processing.

All subsequent steps were carried out in RELION 3.1[43]. Particles were picked using reference based particle picking from initial screening reconstructions. Extracted (700 px) and downsampled (192 px) particles were initially cleaned by 2D classification. In order to remove possible free subunits present in the sample further classification was carried out. Following a consensus refinement at 192 px, the signal from the LSU was subtracted and particles re-extracted centred on the SSU. The SSU sub-particles were then classified in 2D without alignment to remove free LSU particles. This process was repeated with subtractions of the SSU and classification on the LSU sub-particles. Particles were then re-extracted (700 px) and refined to produce an initial reconstruction. Two rounds of CTF refinement and Bayesian polishing were performed. Angles assigned during the following consensus refinement were used in subsequent masked refinements. Masked refinements of flexible regions included the LSU, SSU, SSU head extension, SSU back protuberance and SSU tail. For the L7-L12 stalk region, particles were first subtracted and resampled (256 px) before 3D classification without alignment. Local resolution was calculated using RELION.

### Transcriptome assembly

Raw Illumina HiSeq 2500 forward-reverse pair-ended runs were downloaded as an SRA project from the NCBI website (SRA: SRX710730)[44]. Read pre-processing was carried out using Trim Galore! 0.6.6 [https://github.com/FelixKrueger/TrimGalore], Trimmomatic 0.39[45] and Rcorrector[46]. The reads were then subjected to two separate de novo assembly protocols using (1) rnaSPAdes 3.11.1[47] with assembly parameters automatically selected based on read length, and (2) SOAPdenovo-Trans[48] with a k-mer size of 31.

### Model building and refinement

Each individual masked and local resolution filtered map from masked refinements were fitted to the consensus refinement volume in UCSF-Chimera[49] and combined using the "vop max" command. Previous published mitoribosome structures (PDBID: 6Z1P, 6XYW, 6YWX, 5MRC, 3J9M, and 6HIV) were fitted into the merged map and proteins backbones present cropped from their respective models to be used as starting models. For the rRNA a starting model of the *Arabidopsis thaliana* mitoribosome was used. Fitted proteins were cropped, fitted, mutated and locally refined against the density manually using *Coot* 0.9.5[4,50]. Additional *P. magna* specific proteins were built *de-novo* using *Coot*. Sequences were assigned either though findmysequence.py [https://gitlab.com/gchojnowski/findmysequence] or manually against the assembled and translated transcriptome. The resulting draft model was globally relaxed against the density using Rosetta-FastRelax[51] by first performing a centroid minimization to remove clashes and subsequently three rounds of cartesian FastRelax using a density weight of 150 scored using "elec_dens_fast". The resulting model was refined by one round of "phenix.real_space_refine"[52] with a map weight of 1. Following this ligands Mg (modelled as MO6, octahedrally hydrated Mg²⁺), K and additional ligands were added and the resulting model refined once more using "phenix.real_space_refine" without secondary structure or Ramachandran restraints.

This consensus model was then used as a starting model for tRNA bound states. The mt-LSU and mSSU were first fitted as rigid bodies followed by visual inspection in order to observe any soft body changes in proteins or rRNA. rRNA diagrams were generated using xRNA and finalised using Inkscape [https://inkscape.org/]. All structural graphics were produced using a combination of UCSF-Chimera[49] and ChimeraX[53] and finalised in Inkscape.

### Reporting summary

Further information on research design is available in the Nature Research Reporting Summary linked to this article.

## Data availability

The atomic coordinates generated in this study have been deposited in the Protein Data Bank (PDB) under accession code 8A22 (consensus model), 8APN (classical tRNA) and 8APO (hybrid tRNA). The electron density maps have been deposited in the Electron Microscopy Data Bank (EMDB) under accession codes EMDB-15100 (consensus model), 15576 (classical tRNA) and 15577 (hybrid tRNA).

The atomic coordinates used in this study: 7K00 (*E.coli* ribosome*)*, 6ZM5 (human mitoribosome with OXA1L), 4V8M (*T. brucei* ribosome), 6ZJ3 (*E. gracilis* ribosome).

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

## Acknowledgements

We thank Kristoffer Sahlin for help with transcriptome assembly, and Robert Lee for critically reading and commenting on the manuscript. This work was supported by the Swedish Foundation for Strategic Research (FFL15:0325, ARC19-0051), European Research Council (ERC-2018-StG-805230), Knut and Alice Wallenberg Foundation (2018.0080), EMBO Young Investigator Program. The cryo-EM facility is funded by the Knut and Alice Wallenberg, Family Erling Persson, and Kempe foundations.

## Author contributions

V.T. and A.A. designed the project. V.T. and I.B. prepared the sample for cryo-EM, collected and processed the data and built the model. V.T., I.B., and A.A. analysed the structure and wrote the manuscript.

## Funding

## Competing interests

The authors declare no competing interests.
