## [Peer Review File · Nature Communications]

Structure of a mitochondrial ribosome with fragmented rRNA
in complex with membrane-targeting elementsREVIEWER COMMENTS

Reviewer #1 (Remarks to the Author):

Tobiasson et al solved the structure of the mitochondrial ribosome from the alga *Polytomella magna*. This structure adds to the understanding of mitoribosomal architecture and function which has been studied in several recent publications on the structures of mitochondrial ribosomes from different species. The structure of the *P. magna* mitoribosome reveals how a highly fragmented ribosomal RNA is stabilized in an active mitoribosome. Other noticeable structural features include a non-canonical mt-5S rRNA and a large number of helical repeat proteins involved in rRNA binding. The structure reveals the exit tunnel architecture and an additional unassigned density at the tunnel exit attributed to a membrane protein insertase.

The structure determination is done according to current standards in the field and is well documented in the methods and supplement. The resolution obtained, 2.8 Å for the bulk part of the ribosome compares well to other recent publications and the refinement statistics show that the model is over all of high quality.

While the cryoEM structure determination has been done excellently and the structure warrants publication, the manuscript should be strengthened by an improved presentation and a more in depth analysis of the structural features. The authors should also compare in more detail the recently published structure of the mitochondrial ribosome from the alga *Chlamydomonas reinhardtii* which is only briefly mentioned.

- 1) For better overview, the authors should include a figure of the ribosomal subunits with all proteins labelled. In addition, a gallery with all newly build proteins should be included in the supplement. An extended data figure (e.g. Extended Data Fig 5) should show the *P. magna* rRNA fragment boundaries in a secondary structure diagram for better comparison.
- 2) The authors compared the rRNA fragmentation pattern of *P. magna* only to the *Plasmodium falciparum*. I would suggest to also include the fragmentation patterns of other ribosomes in the comparison. This should include the recently published mitochondrial ribosome from the alga *C. reinhardtii*. rRNA fragmentation also occurs in cytoplasmic ribosomes from kinetoplastids and has recently be described for cytoplasmic ribosomes of the alga *Euglena gracilis* (Matzov et al, *Nucleic Acid Research*, 2020 doi: 10.1093/nar/gkaa89). The fragmentation patterns should also be discussed more in a functional or evolutionary context.
- 3) Mammalian mitochondrial ribosomes harbor a tRNA at the central protuberance (Greber et al *Nature*, 2014). The authors might compare the localization of this tRNA with the *P. magna* 5S rRNA.
- 4) The additional unassigned density at the tunnel exit should be shown in a more detailed figure. It would be interesting to show the contact areas of the unassigned density of the putative insertase and compare it with the known structure of insertase complexes. A superposition of mL45 tail and mL128 would help to evaluate the similarity in binding site on the ribosomal subunit. How does the *P. magna* structure compare to the *C. reinhardtii* structure in which cryo-electron tomography was used to look at the structure of the ribosome on the membrane? Does mL105 has additional domains not build in the structure which could mediate membrane interactions (the supplement is missing information)?
- 5) The authors mention the presence of several HEAT repeat proteins with specific functions to stabilize rRNA fragments. Which other structural motif TPR repeats, ARM motifs etc. are present in the newly build mitochondrial proteins? The supplementary table 2 should be updated and a reference to the sequences should be included in the final version of the table.

Minor points:

Line 20: Should probably read “features a rRNA fragmentation pattern”

Line 21: “phenomena of rRNA” should be replaced with a more appropriate term e.g. “structural features”.

Line 29: The sentence appears to be incomplete (missing a comma or linking word).

Line 51: “In the absence of an accurate mitoribosomal model, no experimental evidence ...” This paragraph should be reworded. Waltz et al presented the structures of the *C. reinhardtii*

mitoribosomal subunits in their publication and reached a similar resolution for the reconstruction of the large ribosomal subunit. In addition, Waltz et al. performed experiments complementing the single particle reconstructions.

Line 53-54: The meaning of “specific elements” is unclear.

Line 56: typo “freshwater alga”

Line 106: “rRNA shares represents” shares appears to be misplaced.

Line 154: “specialities” should be mentioned in more detail.

Line 421: The figure legend should mention the coloring scheme.

Sup Tab. 2: The ORF length of mL105 is missing

Reviewer #2 (Remarks to the Author):

This submission builds on a recent Nature Communications publication (Waltz F et al. Nat Commun. 2021;12(1):021-27200) that described a cryo-EM structure for the mitoribosome of the green alga *Chlamydomonas reinhardtii*, which also contains fragmented SSU and LSYU rRNAs. The rRNA fragmentation pattern in the *Polytomella parva* mitoribosome (a cryo-EM structure for which is presented here) is essentially the same as the *Chlamydomonas* mitoribosome fragmentation pattern.

Because the cryo-EM resolution is higher in the present case, the authors are able to refine the 3-D structure by proposing additional protein-rRNA interactions as well as identifying additional novel ribosomal proteins that were missed in the *Chlamydomonas* study. The the higher-resolution structure also provides additional/novel structure-function insights. To the extent I'm able to evaluate the cryo-EM determination, it appears competently done and comprehensive, so I have no major criticisms of the work in that regard. I do, however, have a few comments regarding the text, as well as a few minor corrections.

Comments:

1. lines 20-21 and elsewhere: The authors suggest that the mt-rRNA fragmentation pattern in green algae “has similarities with apicomplexan parasites”. It's not clear what the authors mean to imply by this statement, considering that the fragmentation pattern in apicomplexans is much more extensive than in green algae (e.g., Fig. 1). Because reduced mt-rRNAs retain a conserved functional core having a very similar 2ary structure, it is hardly surprising that fragmented mt-rRNAs would reconstitute a very similar structure. Hence, I would argue that the similarities the authors point out could be purely fortuitous, without any obvious evolutionary or functional significance.

2. line 66: “26 newly identified” ribosomal proteins. Which ones are these in Table 2 (please specify in the table)? Are these all newly identified in this study, or were some also previously identified in the *Chlamydomonas* one? Presumably the entries designated “UNK” in Table 2 are the HEAT-repeat proteins; if so, Table 2 should explicitly indicate this.

3. lines 96-107: The authors fail to credit the *Chlamydomonas* study cited above for identification one of the LSU rRNA fragments as a highly divergent 5S rRNA. How do the authors' results on this molecule compare with those of the *Chlamydomonas* one?

4. lines 98 and 106-107: Divergent mitochondrial 5S rRNAs have previously been reported (e.g., Bullerwell CE et al. Abundant 5S rRNA-like transcripts encoded by the mitochondrial genome in Amoebozoa. Eukaryot Cell. 2010;9(5):762-73), so I'd be cautious in claiming that the green algal one is “the most diverged version of the 5S molecule in a ribosome”.

Minor issues:

line 18: "Mitoribosomes" is misspelled; "alga" should be "algae"

line 19: "an algal mitoribosome": the authors should include the name of the alga in question in the Abstract

line 106: delete "shares"

line 119: "unwanted" would be a better choice than "unsought" here

line 123: "accumulation" is misspelled

line 124: "by rapid" rather than "by a rapid"; either "encoding" or "coding for", not "encoding for"

lines 161-162: "tangled with convergent evolution" is rather non-scientific, I'd suggest a change in wording here

We thank the Reviewers for taking the time to provide constructive suggestions on how to improve the study, data interpretation and its presentation. We addressed all the requests and followed the valuable suggestions. Here is the outline of the additions and changes to the revised manuscript.

We added analyses and figures showing the following: ribosomal subunits with all proteins labelled, a gallery with newly built proteins, comparison between the mt-5S rRNA and CP tRNA, comparison of the insertase density between *P. magna* map and human mitoribosome-Oxa1, comparison of the rRNA fragmentation patterns with cytoplasmic ribosomes of kinetoplastids and *Euglena gracilis*.

In the text, we further expand on the themes of the rRNA fragmentation patterns in an evolutionary context, and the 5S/tRNA, referring to the relevant literature. Finally, a short summarising section has been added to conclude the manuscript. Together, it substantially strengthened the study, and we would like to thank the Reviewers for the useful comments.

Below is the point-by-point response.

Reviewer #1:

1) For better overview, the authors should include a figure of the ribosomal subunits with all proteins labelled. In addition, a gallery with all newly build proteins should be included in the supplement. An extended data figure (e.g. Extended Data Fig 5) should show the *P. magna* rRNA fragment boundaries in a secondary structure diagram for better comparison.

- We followed the suggestions and included a figure of the ribosomal subunits with all the protein labels (Supplementary Information Fig. 3), a gallery with all newly built proteins (Supplementary Information Fig. 4) and added *P. magna* rRNA fragment boundaries in the secondary structure diagram (now Supplementary Information Fig. 6).

2) The authors compared the rRNA fragmentation pattern of *P. magna* only to the *Plasmodium falciparum*. I would suggest to also include the fragmentation patterns of other ribosomes in the comparison. This should include the recently published mitochondrial ribosome from the alga *C. reinhardtii*. rRNA fragmentation also occurs in cytoplasmic ribosomes from kinetoplastids and has recently be described for cytoplasmic ribosomes of the alga *Euglena gracilis* (Matzov et al, Nucleic Acid Research, 2020 doi: 10.1093/nar/gkaa89). The fragmentation patterns should also be discussed more in a functional or evolutionary context.

- We added the requested analysis with discussion and compared the rRNA fragmentation patterns also with mitoribosomes of *C. reinhardtii*, cytoplasmic ribosomes of kinetoplastids and the recently described ribosomes of the alga *Euglena gracilis*. The data is presented in Fig. 4, and the analysis suggests two possible evolutionary scenarios: 1) the steps in the trajectory of rRNA fragmentation are convergent across large evolutionary distances due to an adaption to structural constraints that are intrinsic to the basic ribosomal architecture; or 2) the rRNA fragmentation pattern represents a record of ribosomal history from the evolution of the translational system. References to the relevant work by Gray and Williams labs have been added in the introduction and discussion sections.

3) Mammalian mitochondrial ribosomes harbor a tRNA at the central protuberance (Greber et al Nature, 2014). The authors might compare the localization of this tRNA with the *P. magna* 5S rRNA.

- We added the analysis of the 5S/tRNA on lines 178-186 and in Fig. 5. As requested, it shows side by side comparison and suggests how the 5S rRNA can be removed without affecting the ribosomal structure and function.

4) The additional unassigned density at the tunnel exit should be shown in a more detailed figure. It would be interesting to show the contact areas of the unassigned density of the putative insertase and compare it with the known structure of insertase complexes. A superposition of mL45 tail and mL128 would help to evaluate the similarity in binding site on the ribosomal subunit. How does the *P. magna* structure compare to the *C. reinhardtii* structure in which cryo-electron tomography was used to look at the structure of the ribosome on the membrane? Does mL105 has additional domains not build in the structure which could mediate membrane interactions (the supplement is missing information)?

- We added this information in Fig. 6 that shows comparison with the known structure of the insertase complex. A superposition of mL45 tail and mL128 is presented in panel b and discussed more in detail. We added a statement on lines 198-199 that the density in the *P. magna* structure correlates with the inner membrane plane reported by tomography of *C. reinhardtii* mitoribosomal complexes. The missing information has been added to the table.

5) The authors mention the presence of several HEAT repeat proteins with specific functions to stabilize rRNA fragments. Which other structural motif TPR repeats, ARM motifs etc. are present in the newly build mitochondrial proteins? The supplementary table 2 should be updated and a reference to the sequences should be included in the final version of the table.

- No other motifs were found, which we now state in the text.

Minor points:

Line 20: Should probably read “features a rRNA fragmentation pattern”

- Added.

Line 21: “phenomena of rRNA” should be replaced with a more appropriate term e.g. “structural features”.

- Replaced.

Line 29: The sentence appears to be incomplete (missing a comma or linking word).

- Added “and”.

Line 51: “In the absence of an accurate mitoribosomal model, no experimental evidence ...”
This paragraph should be reworded. Waltz et al presented the structures of the *C. reinhardtii* mitoribosomal subunits in their publication and reached a similar resolution for the reconstruction of the large ribosomal subunit. In addition, Waltz et al. performed experiments complementing the single particle reconstructions.

- We apologise for poorly crediting the study by Waltz et al in the first version. We removed the sentence and added citations throughout the manuscript on lines 57, 127, 166, 172 and particularly, in the context of tomography.

Line 53-54: The meaning of “specific elements” is unclear.

- Removed.

Line 56: typo “freshwater alga”

- Corrected.

Line 106: “rRNA shares represents” shares appears to be misplaced.

- Deleted.

Line 154: “specialities” should be mentioned in more detail.

- Might represent a solution to maintain this feature of the mitoribosomal exit tunnel.

Line 421: The figure legend should mention the coloring scheme.

- Added.

Sup Tab. 2: The ORF length of mL105 is missing

- Added.

Reviewer #2

1. lines 20-21 and elsewhere: The authors suggest that the mt-rRNA fragmentation pattern in green algae “has similarities with apicomplexan parasites”. It’s not clear what the authors mean to imply by this statement, considering that the fragmentation pattern in apicomplexans is much more extensive than in green algae (e.g., Fig. 1). Because reduced mt-rRNAs retain a conserved functional core having a very similar 2ary structure, it is hardly surprising that

fragmented mt-rRNAs would reconstitute a very similar structure. Hence, I would argue that the similarities the authors point out could be purely fortuitous, without any obvious evolutionary or functional significance.

- We removed this sentence that might sound misleading and expanded on the theme of the fragmentation pattern on page 8 of the revised manuscript and figure 4, also following the Reviewer #1 comment 2. We hope this new version clarifies the raised issue.

2. line 66: “26 newly identified” ribosomal proteins. Which ones are these in Table 2 (please specify in the table)? Are these all newly identified in this study, or were some also previously identified in the *Chlamydomonas* one? Presumably the entries designated “UNK” in Table 2 are the HEAT-repeat proteins; if so, Table 2 should explicitly indicate this.

- We added this information in Supplementary Fig .4.

3. lines 96-107: The authors fail to credit the *Chlamydomonas* study cited above for identification one of the LSU rRNA fragments as a highly divergent 5S rRNA. How do the authors’ results on this molecule compare with those of the *Chlamydomonas* one?

- We apologise for poorly crediting the *Chlamydomonas* study with respect to the 5S rRNA. It has now been added on line 166. The structures are similar, and to provide a new insight we now added an additional analysis in the evolutionary context on lines 178-186.

4. lines 98 and 106-107: Divergent mitochondrial 5S rRNAs have previously been reported (e.g., Bullerwell CE et al. Abundant 5S rRNA-like transcripts encoded by the mitochondrial genome in Amoebozoa. *Eukaryot Cell*. 2010;9(5):762-73), so I’d be cautious in claiming that the green algal one is “the most diverged version of the 5S molecule in a ribosome”.

- We removed the claim and added the citation.

Minor issues:

line 18: “Mitoribosomes” is misspelled; “alga” should be “algae”

- Fixed.

line 19: “an algal mitoribosome”: the authors should include the name of the alga in question in the Abstract

- Added.

line 106: delete “shares”

- Deleted.

line 119: “unwanted” would be a better choice than “unsought” here

- Changed.

line 123: “accumulation” is misspelled

- Corrected.

line 124: “by rapid” rather than “by a rapid”; either “encoding” or “coding for”, not “encoding for”

- Corrected.

lines 161-162: “tangled with convergent evolution” is rather non-scientific, I’d suggest a change in wording here

- Reworded to “... *evolve in a way of convergent evolution.*”

REVIEWER COMMENTS

Reviewer #1 (Remarks to the Author):

The authors have addressed my concerns and the manuscript is now suitable for publication with minor corrections.

line 134 “fragmentation convergence” I would suggest to change to “convergence of rRNA fragmentation patterns.”

line 137 The authors should clarify the meaning of “parsimonious”.

line 141 “record of ribosomal history from the evolution of the translational system”. For readers less familiar with the topic it should be clarified that the fragmented rRNA of contemporary ribosomal subunits most likely evolved from ancestors with covalently continuous rRNA. (However, a primordial ribosome may indeed have contained fragmented rRNA as discussed by the authors)

Journal names and abbreviations in the references are not consistent. Please check the formatting and add missing information:

line 418 28. Gray, M. Gopalan, V. Piece by piece: Building a ribozyme. *J. Biol. Chem.* 295(8) (2020)
Page number missing

line 412 25. Matzov, D. et al. Cryo-EM structure of the highly atypical cytoplasmic ribosome of *Euglena gracilis*. (2020). doi:10.2210/pdb6zj3/pdb

The reference contains the doi of the PDB entry. Please use the doi of the publication in *Nucleic Acids Res.*

Reviewer #1

line 134 “fragmentation convergence” I would suggest to change to “convergence of rRNA fragmentation patterns.”

- *Changed.*

line 137 The authors should clarify the meaning of “parsimonious”.

- *Changed to “compatible with an assembly path that does not necessarily require a cotranscriptional folding”.*

line 141 “record of ribosomal history from the evolution of the translational system”. For readers less familiar with the topic it should be clarified that the fragmented rRNA of contemporary ribosomal subunits most likely evolved from ancestors with covalently continuous rRNA. (However, a primordial ribosome may indeed have contained fragmented rRNA as discussed by the authors)

- *Added “Although, the fragmented rRNA of contemporary ribosomal subunits most likely evolved from ancestors with covalently continuous rRNA.”*

Journal names and abbreviations in the references are not consistent. Please check the formatting and add missing information:

line 418 28. Gray, M. Gopalan, V. Piece by piece: Building a ribozyme. J. Biol. Chem. 295(8) (2020)

Page number missing

- *Fixed.*

line 412 25. Matzov, D. et al. Cryo-EM structure of the highly atypical cytoplasmic ribosome of *Euglena gracilis*. (2020). doi:10.2210/pdb6zj3/pdb

The reference contains the doi of the PDB entry. Please use the doi of the publication in Nucleic Acids Res.

- *Fixed.*